# The Providence Mutation (βK82D) in Human Hemoglobin Substantially Reduces βCysteine 93 Oxidation and Oxidative Stress in Endothelial Cells

**DOI:** 10.3390/ijms21249453

**Published:** 2020-12-11

**Authors:** Sirsendu Jana, Michael Brad Strader, Abdu I. Alayash

**Affiliations:** Laboratory of Biochemistry and Vascular Biology, Center for Biologics Evaluation and Research, Food and Drug Administration (FDA), Silver Spring, MD 20993, USA; sirsendu.jana@fda.hhs.gov (S.J.); michael.strader@fda.hhs.gov (M.B.S.)

**Keywords:** hemoglobin providence, βCys93 oxidation, pulmonary endothelium, glycolysis

## Abstract

The highly toxic oxidative transformation of hemoglobin (Hb) to the ferryl state (HbFe^4+^) is known to occur in both in vitro and in vivo settings. We recently constructed oxidatively stable human Hbs, based on the Hb Providence (βK82D) mutation in sickle cell Hb (βE6V/βK82D) and in a recombinant crosslinked Hb (rHb0.1/βK82D). Using High Resolution Accurate Mass (HRAM) mass spectrometry, we first quantified the degree of irreversible oxidation of βCys93 in these proteins, induced by hydrogen peroxide (H_2_O_2_), and compared it to their respective controls (HbA and HbS). Both Hbs containing the βK82D mutation showed considerably less cysteic acid formation, a byproduct of cysteine irreversible oxidation. Next, we performed a novel study aimed at exploring the impact of introducing βK82D containing Hbs on vascular endothelial redox homeostasis and energy metabolism. Incubation of the mutants carrying βK82D with endothelial cells resulted in altered bioenergetic function, by improving basal cellular glycolysis and glycolytic capacity. Treatment of cells with Hb variants containing βK82D resulted in lower heme oxygenase-1 and ferritin expressions, compared to native Hbs. We conclude that the presence of βK82D confers oxidative stability to Hb and adds significant resistance to oxidative toxicity. Therefore, we propose that βK82D is a potential gene-editing target in the treatment of sickle cell disease and in the design of safe and effective oxygen therapeutics.

## 1. Introduction

Heme iron oxidation can have serious biological consequences as it impacts the ability of Hb to deliver oxygen, which can jeopardize its safe use in transfusion medicine [1]. Spontaneous oxidation of the heme iron, also known as autoxidation, occurs within red blood cells (RBCs), and at much higher rates when Hb is found outside circulating RBCs. Hb oxidation in RBCs occurs despite the presence of several antioxidant proteins and enzymes that are designed to suppress reactive oxygen species (ROS), resulting from the oxidation of the heme iron [2,3]. However, when RBCs are stored or pathogen inactivated for transfusion purposes, the function of these antioxidant mechanisms can be compromised [4]. Likewise, RBC genetic disorders and hemoglobinopathies also accelerate Hb oxidation, which often leads to premature hemolysis and heme loss [5]. As a result, understanding the underlying mechanisms of Hb oxidation is therefore critical for designing methods aimed at potentially reducing or preventing its complications. 

The last three to four decades witnessed considerable efforts to develop commercial products using free Hb oxygen-based carrier (HBOCs) therapeutics, also known as blood substitutes. The initial HBOC manufacturing process involved extracting Hb from human or animal blood, followed by extensive purification and crosslinking into tetramers or other polymers, to extend circulation time in recipients after infusion [6]. Unfortunately, adverse side effects caused by the infusion of HBOCs in patients led to the termination of several clinical trials in the USA and elsewhere [7]. Uncontrolled heme iron oxidation of HBOCs, coupled with vascular complications due to the reaction of free Hb with nitric oxide (NO), were among the major contributors to the safety issues with these therapeutics [8]. 

Oxidation of heme iron not only compromises the ability of Hb to carry and unload oxygen, it can also be a source of heme and other toxic oxidative intermediates. Rapid oxidation to ferric (met) Hb during transfusion of HBOCs was documented in several preclinical and clinical settings [9]. The level of metHb measured in some of these animal experiments varied from 10 to 65% [10], after infusion of HBOCs and higher metHb levels (~50%) in humans. To address this, a powerful reducing agent of Hb, ascorbate, was successfully used in one reported case to control HBOC oxidation in humans [11]. These applications of HBOCs provided a unique opportunity to study free Hb’s oxidative pathways closely outside the cellular environment [12].

Hydrogen peroxide (H_2_O_2_) reacts with both ferrous (oxy Hb) and ferric (met Hb) forms of Hb, and this reaction results in the formation of highly reactive species, e.g., ferryl Hb (HbFe^4+^), together with a protein radical (HbFe^4+^). The radical is formed when the reaction starts with met Hb and is stabilized on the porphyrin or nearby amino acids, leading to the formation of the peroxidase compound II heme state [13]. This “unharnessed” radical, unlike true peroxidases, escapes from the porphyrin ring to other amino acid side chains, including βCys93, which then reacts with oxygen to form cysteic acid. These internal reactions appear to result in the modification of heme, its subsequent attachment to nearby amino acids, and the irreversible oxidation of reactive amino acids, particularly βCys93, which promotes the unfolding and dissociation of Hb [1].

We previously discovered that a naturally occurring mutant Hb Providence (βLys82→Asp) (βK82D), was much more resistant to degradation by H_2_O_2_ than normal human HbA [14]. Based on this finding, we then engineered this mutation into a genetically cross-linked Hb tetramer (rHb0.1/βK82D) and subsequently demonstrated that the βK82D mutation conferred more resistance to degradation by H_2_O_2_, by markedly inhibiting oxidation of the β93 cysteine side chain [15]. Next, we tested this extraordinary stability of the βK82D mutation in another Hb model system known for its oxidative instability, e.g., sickle cell Hb (HbS). The HbS (βE6V) mutation is known to oxidize faster than normal HbA and remains longer in a highly oxidizing ferryl form (HbFe^4+^), which then targets the “hotspot” amino acids including βCys93 [16]. We found that the (βE6V/βK82D) form of Hb added a significant oxidative resistance to βE6V when challenged with H_2_O_2_, in addition to a dramatic improvement in the delay times and polymerization of βE6V [16].

In this study, we investigated the impact of sickle cell and crosslinked human Hbs containing the βK82D mutation and their respective controls (βK82D, HbA, and HbS) on vascular endothelial cellular metabolism and oxidative toxicity, using human pulmonary artery endothelial cells (HPAECs). We found a strong correlation between the suppression of βCys93 oxidation in Hb proteins containing βK82D mutation and the recovery of endothelial glycolytic capacity, along with diminished heme oxygenase-1 (HO-1) expression, which is caused by the toxic release of heme. 

## 2. Results

### 2.1. Pseudoperoxidase Activity of the Providence Mutation 

Redox transition of ferrous Hb to a ferric/ferryl state can be monitored spectrophotometrically and the accompanying changes can be followed in real-time (Figure 1A,B). Under oxidative stress conditions, i.e., in the presence of H_2_O_2_ (up to 30-fold over heme), the spectrum of the highly purified ferrous HbA_0_ (two peaks at 541 and 577 nm) is rapidly transformed to the ferryl species (Figure 1A). The ferryl Hb with its signature UV/Vis spectrum (two major peaks at 545 and 580 nm and a flattened region between 500 and 600 nm) remains in solution all the way to the end of the experiment (1 h), and ultimately self-destructs, leading to unfolding of the protein and heme loss [13]. Treatment of the of K82D variant with the same levels of H_2_O_2_ results in the formation of a transitional ferryl species, which is rapidly autoreduced back to the ferric protein (peaks at 510, 550, and at 630 nm), clearly demonstrating an effective pseudoperoxidative activity (Figure 1B). 

These reactions can be analyzed based on a simple model (pseudoperoxidative cycle) described earlier [13] (Figure 1C). The reaction of H_2_O_2_ with both ferrous and ferric forms of Hb results in the formation of a highly reactive species, e.g., ferryl Hb (HbFe^4+^), together with a protein radical (·HbFe^4+^) (when the reaction starts with met/ferric Hb) (k_1_). Once its formed, ferryl heme is autoreduced back to the ferric form (k_2_), and in the presence of access H_2_O_2_, the ferric is transformed to a ferryl heme (k_3_), thus, completing a pseudoperoxidative cycle [13] (Figure 1C). 

Based on a recent crystal structure analysis, the extraordinary oxidative stability of the βK82D mutation was attributed to changes in reactivity of the βCys93 side chain, which might be due to either indirect electrostatic effects (replacement of the positive Lys by a negative Asp acid) or alterations in the dynamics in the vicinity of Lys82 [16] (Figure 1D). In native HbA, the Lys82 side chain is located far away from the heme group and the ε-amino N atom is roughly 13 Å away from the βCys93 sulfur atom (Figure 1D). The amino acid side chains and the heme group are superimposable, with the only significant differences being the loss of the Lys82 (positive charge), and the appearance of the Asp82 carboxyl group (negative charge), which is roughly 9.5 Å away from the βCys93 sulfur atom. As a result, these electrostatic effects on βCys93 occur indirectly, since these regions are ≥9 Å away from the sulfur atom [16] (Figure 1D). 

### 2.2. MS Analysis of βK82D Mutants Treated with Hydrogen Peroxide

We previously used 5,5-Dimethyl-1-pyrroline N-oxide (DMPO) labeling studies and LC/MS/MS analysis to show that Hb toxicity is linked to irreversible βC93 oxidation to cysteic acid, by virtue of ferryl ions and globin radicals [18]. While other intermediates might occur, comparing βC93 oxidation to cysteic acid is an effective method for evaluating Hb oxidative stability, relative to HbA. For each βC93 containing peptide charge state identified by the Mascot database searches, the extracted ion chromatograms (XICs) were generated from the most abundant monoisotopic peak of each peptide isotopic profile (in a 2-step process (shown in Figure 2) and the resulting ratio differences were compared. As βCys93 exists in either the oxidized or unoxidized form, after treatment with H_2_O_2_, the relative abundance of both isoforms were calculated based on the sum of the XIC peak area from all the detectable βCys93 peptides. As shown in Table 1, H_2_O_2_ addition at 2.5, 5.0, and 10-fold excess led to increased βCys93 oxidation for both HbA and HbS. However, there was 1.5 to 2.0-fold increases at the low and middle [H_2_O_2_] and 1.0-fold at extreme [H_2_O_2_], in HbS over HbA [16]. Overall, these data support many of our past and current lab studies that show HbS is oxidatively less stable than wild-type HbA [18,19,20].

Additionally, the data in Table 1 show that the βK82D mutation confers markedly more resistance to oxidation by H_2_O_2_, when compared to HbA and HbS, respectively. There was a 3 to 6-fold reduction and 4 to 7.0-fold decrease in βCys93 in HbA and HbS at low, medium, and high H_2_O_2_ concentrations. Construction of βK82D in HbS (βE6V/βK82D), reduced βCys93 oxidation by 2.7-fold and 3.0-fold in this mutant at low and high H_2_O_2_, whereas the same mutation in Providence (βK82D) crosslinked rHb (rHb0.1/βK82D) reduced oxidation of βCys93 below the detection levels, under the same peroxide levels. These quantitative results collectively showed that βK82D provides antioxidant protection by reducing H_2_O_2_-induced oxidation damage to the proteins. 

### 2.3. Endothelial Oxidative Stress Induced by Hemoglobin Mutants

We and others have shown that cell-free Hb, especially oxidatively unstable HbS, exerts oxidative toxicity on cultured endothelial and other cells, during in vitro incubation through the release of heme, as result of oxidative changes and unfolding of the protein [18,19,21,22]. To investigate the effect of βK82D mutation on the pulmonary endothelial stress response, we incubated the Hb proteins carrying βK82D at equimolar concentration (100 µM) with cultured HPAECs for 24 h. HO-1 and ferritin proteins are the most potent indicators of heme release from Hbs and subsequent iron load within the cellular compartment, respectively [23,24]. Therefore, we first monitored the expression of the levels of these proteins in HPAECs. Exposure to either HbA or HbS (βV6E) caused a robust increase in HO-1 and ferritin expression in HPAECs (Figure 3A,C). However, HbS induced significantly higher levels of HO-1 over the corresponding HbA (Figure 3A,B). On the other hand, the βK82D mutant at equimolar concentration also caused induction of HO-1 and ferritin proteins, but at a significantly lower degree than HbA and HbS. In contrast, rHb0.1/βK82D caused no expression of ferritin and a very low level of HO-1 expression. This is consistent with the negligible levels of βCys93 oxidation in this protein (Table 1). Surprisingly, introduction of βK82D in oxidatively susceptible βE6V/βK82D caused a significant reduction in the expression of both HO-1 and ferritin compared to native HbS (Figure 3A–C). To verify this finding, we used an orthogonal approach for a visual comparison of the expression pattern of HO-1 induced by HbS and HbS carrying the βK82D mutation. As seen in our immunoblotting experiment, HbS induced stronger expression of HO-1 (indicated by red fluorescence) than corresponding HbA in HPAECs (Figure 4). Whereas, βE6V/βK82D caused much lower levels of HO-1 expression than βE6V (Figure 4).

We also monitored TLR4 expression since, TLR4 is the primary site of activation for signaling cascade mediated by free heme [22]. We found slightly higher levels of TLR4 in both HbA and HbS-treated HPAECs, compared to the untreated controls. However, βE6V/K82D failed to show any further improvement over native HbS under similar incubation conditions. Higher expression of endothelial adhesion molecules (VCAM-1 and ICAM-1) were documented before as a response to the presence of Hb/heme [25]. Therefore, we analyzed HPAEC lysates following treatment with different Hb proteins for VCAM-1 expression. No noticeable difference in VCAM-1 expressions were observed with any of the mutant Hbs used in this study (Figure 3C).

### 2.4. Impact of Providence Mutation on the Bioenergetic Impairment in HPAECs 

To assess the energy utilization in Hb-treated HPAECs, we monitored mitochondrial bioenergetics and glycolytic proton flux by XF assay using the Seahorse XF24 Extracellular Flux Analyzer (Agilent, Santa Clara, CA, USA) in real time. Oxygen consumption rate (OCR) was obtained as a direct indicator of mitochondrial respiration in HPAEC. Figure 5A shows the bioenergetic profiles of different Hb-treated HPAECs, by plotting OCR data obtained in real time. Basal mitochondrial respiration before addition of oligomycin and maximal respiration in uncoupled state (following FCCP injection) were calculated from the OCR plot (Figure 5B). Basal OCR was not affected by any of the Hb proteins. Maximal respiration is a result of a collapse of the proton gradient induced by the uncoupling agent FCCP, where an uninhibited flow of electrons occurs through the electron transport chain complexes and allows maximum oxygen consumption by cytochrome c oxidase (complex IV). Exposure to HbA and βK82D did not cause any significant changes in maximal respiration compared to the untreated controls (Figure 5B). However, βE6V treated cells showed mild but non-significant uncoupling over the untreated control cells, perhaps mediated by TLR4 activation and induction of HO-1 expression [21]. In contrast, this uncoupling effect was not seen in HPAECs treated with the HbS carrying βE6V/βK82D mutation. 

In a similar experimental setup, the glycolytic rate was assessed by measuring ECAR in different Hb-treated HPAECs (Figure 5C). Basal glycolysis induced by the addition of glucose was not affected by any of the Hb proteins used in the study, except HbS (Figure 5D). βE6V caused a ~30% loss in basal glycolysis, compared to the untreated controls. More pronounced inhibitory effects of βE6V were seen in glycolytic capacity after addition of oligomycin, where βE6V caused more than a 40% loss in glycolytic capacity over the untreated control cells (Figure 5D). In contrast, HbA, Hb Prov, and rHb0.1 Prov had no noticeable effect on glycolytic capacity. Alternatively, HbS Prov (βE6V/βK82D) partially reversed the HbS-induced loss of glycolytic capacity (Figure 5D). 

## 3. Discussion

Our laboratory recently investigated several naturally occurring mutant human Hbs that were oxidatively stable and were able to withstand extreme oxidative pressures [15]. Mutant Hbs are described as “Experiments in Nature”, as they provide an experimental platform to investigate why some mutants evolutionarily evolved into oxidatively stable molecules, while others develop into circulatory disorders [26,27]. In most cases, the mutation involves a single, or at the most several amino acids in key functional areas of the Hb molecule. 

To control Hb’s oxidative side reactions, we focused primarily on the reduction of ferryl Hb directly, by genetically re-engineering key stabilizing amino acid(s) in the protein [17,28]. In particular, we studied how an evolutionary stable Providence mutation (βK82D) can provide oxidative resistance when constructed in oxidatively unstable crosslinked and sickle cell Hbs, by reducing the ferryl heme content and subsequent irreversible oxidation of βCys93 to cysteic acid [15,16]. In addition to its well-established role in allosteric mechanisms, βCys93 is involved in the transport of nitric oxide (NO) and detoxification of superoxide ions (O_2_^−^) [1]. Several studies from our laboratory and others showed that βCys93 (positioned on the surface near an F helix located at the β = β subunit interface) is an endpoint for free-radical-induced Hb oxidation, which occurs as a consequence of oxygen binding and concomitant H_2_O_2_ (and O_2_^−^) production [13,17,29]. βCys93 is readily and irreversibly oxidized in the presence of a mild oxidant, H_2_O_2_ to cysteic acid, which leads to the destabilization of Hb, resulting in improper protein folding and the loss of heme. Oxidized βCys93 is therefore a useful reporter on the oxidative status of Hb in RBCs intended for transfusion, or within RBCs, from patients with hemoglobinopathies [1]. Accordingly, site-specific mutation of redox active amino acid(s) to reduce the ferryl heme, or direct chemical modifications that can shield βCys93, were proposed to improve oxidative resistance of Hb and might offer a protective therapeutic strategy [28,30].

We used two constructs containing the βK82D mutation, a sickle cell Hb and a wild-type crosslinked Hb construct. Our aim was to correlate the oxidative resistance of these βK82D containing mutants (as indexed by βCys93 oxidation) on oxygenation, and the overall redox state of the cultured endothelial cells. As seen in Table 1, the presence of the βK82D substitution provided a considerable oxidative protection to both sickle cell and more so to the crosslinked Hb, by minimizing the levels of irreversible oxidation of βCys93. Oxidation of βCys93 to cysteic acid is known to perturb the extensive network of hydrogen bonding and salt bridges at the interface between the β2 FG corner. The substitution of the native βLys82 for Asp82 (located ~18.3 Å away from the β-heme) is not known to be engaged in electron transfer with the heme [16]. However, the higher oxidative stability caused by the βK82D mutation was attributed to changes in reactivity of the βCys93 side chain, which might be due to either indirect electrostatic effects or alterations to the local dynamics of the protein structure [15]. Hydrogen-peroxide-dependent oxidation reactions of the oxy form of the βK82D mutant might also reveal additional mechanistic insights into differences in the kinetics of ferryl heme formation and reduction and the potential role of Asp82 in the protein redox transitions. 

The impact of a βK82D mutation on tissue oxygenation and mitochondrial function is not known. Anecdotal reports on patients with βK82D mutations showed, for example in one case, mild anemia and erythropoiesis, likely due to the alteration in the oxygen-binding properties of patients’ blood [31]. Oxygen affinity in red cells from another patient with the βK82D mutation was also reported to be higher than normal subjects (i.e., P_50_ = 21.1 mm Hg vs., control P_50_ = 29.0 mm Hg) [32]. In another reported case of an SCD patient with a coexisting βK82D mutation, suffered from only mild symptoms from this disease [33]. The impact of the βK82D mutation on the course and the severity of the disease in this SCD patient might not be entirely due to a small left shift in the oxygen equilibrium curve (smaller P50), but might actually be due to an overall improvement in patient’s resistance to oxidative stress [34].

Heme released from oxidized Hb proteins, especially from HbS (due to its higher autoxidation rate over HbA), is known to cause an inflammatory response, vasoconstriction, and other vascular complications, e.g., endothelial dysfunction [21,22]. In hemolytic conditions like SCD and β-thalassemia, lung endothelial cell-dysfunction is considered as one of the major factors leading to pulmonary arterial hypertension (PAH) [35,36]. Vascular injury and inflammation initiated by the activation of TLR4 of the innate immune system is triggered by DAMP (damage-associated molecular pattern) molecules, including heme [22,35]. Using an in vitro experimental set-up exploiting human pulmonary endothelial cells, we previously demonstrated the cellular oxidative stress response and impact on energy homeostasis inflicted by HbS and other mutants [21]. Endothelial cells are generally more dependent on glycolysis than mitochondrial oxidative phosphorylation as their source of ATP, but many studies showed the importance of mitochondrial signal-transduction and bioenergetics in preserving normal vascular endothelial function [37,38]. Although basal mitochondrial respiration was not affected, only HbS among all other Hb mutants caused more pronounced uncoupling over untreated healthy HPAECs. This effect can be attributed to heme released from oxidatively unstable HbS causing activation of TLR4, inducing HO-1, and thus producing carbon monoxide (CO) (byproduct of heme degradation), since, activated TLR4 and CO are both known to cause uncoupling of mitochondrial respiration [39,40,41]. 

It is possible that introduction of βK82D mutation into HbS protein also effectively blunted the HbS-mediated uncoupling effect, by substantially lowering heme mediated HO-1 and ferritin expression. Moreover, this heme or CO-mediated mitochondrial uncoupling could have another impact on the vascular endothelial cells, i.e., exhaustive loss of glycolytic capacity, leading to inhibition of glycolysis, as seen in some cancer cells [42]. Earlier, we also showed that uncoupled respiration in endothelial mitochondria by HbS was associated with concomitant increase in lipid peroxidation and mitochondrial oxygen radical production [21]. Since endothelial glycolysis can be impacted by reactive oxygen radicals and other oxidation end-products, therefore, the HbS-mediated loss of glycolytic capacity observed in our study could possibly be contributed by ROS or lipid peroxidation end products, generated by dysfunctional mitochondrial bioenergetics [43]. In this context, failure to show any glycolytic inhibition by HbS carrying βK82D mutation further supports the oxidative stability of the mutation. Rates for heme loss from chemically and genetically crosslinked Hbs investigated here were previously shown to be several magnitudes lower than heme loss rates determined for other modified and unmodified Hbs [44]. Supported by the mass-spectrometric results (βCys93 oxidation) and (HO-1 expression) in cells, rHb0.1/K82D was not surprisingly the least damaging to cells among the βK82D mutants, which might explain the additional protection afforded by this protein to cells.

In summary, we set out to examine the mechanistic link between irreversible oxidation of a key amino acid in the β subunit of Hb, Cys93, oxidative toxicity, cellular redox homeostasis, and energy metabolism, in the cultured endothelial cells. Moreover, we tested whether the construction of a naturally occurring antioxidative mutation in oxidatively unstable proteins can reverse the oxidative side reactions of Hb and subsequently the cellular and subcellular oxidative responses. Oxidative modifications triggered by βCys93 oxidation to cysteic acid and subsequent heme loss from normal and sickle cell Hbs caused impairment of pulmonary endothelial glycolysis, which were effectively blunted by βK82D mutation into HbS and crosslinked human Hb. Construction of βK82D mutation in otherwise oxidatively susceptible Hbs might therefore provide an intervention strategy that can be applied in gene-editing therapies for sickle cell disease, and in the design of safe and effective oxygen therapeutics.

## 4. Materials and Methods

### 4.1. Recombinant Hemoglobins

Recombinant mutant Hb proteins were expressed using the pSGE1702 plasmid 35 and were kind gifts from Dr. JS Olson (Rice University). All amino acid substitutions were made using the Stratagene PCR-based QuikChangeTM site-directed mutagenesis kit (Stratagene, La Jolla, CA, USA). The *E. coli* strain JM109 was used for the mutagenesis and expression of Hb proteins. Large-scale Hb expression was done in a Biostat C20 bioreactor (B Braun Biotech International, Melsungen, Germany), as described in [15,45]. The full biochemical and biophysical characterization of mutant Hbs investigated here were published in [15,16].

### 4.2. Oxidation Reactions of Mutant Hemoglobins

Autoxidation as well as chemically-induced oxidation experiments were carried out by aerobically incubating Hb samples in 50 mM phosphate buffer, pH 7.4, at 37 °C for 24 h, or by adding bolus amount of H_2_O_2_. Absorbance changes for both experiments in the range of 350–700 nm due to oxidation of oxyHbs (60 µM in heme) were recorded in a temperature-controlled photodiode array spectrophotometer (HP 8453). Multicomponent analysis was used to calculate the oxyHb and metHb concentrations based on the extinction coefficients of each species [46]. The spectra of reaction mixtures of ferrous Hbs (60 µM) with molar excess H_2_O_2_ (150, 300, and 600 µM) were monitored every 20 s, for 30 min, in a photodiode array spectrophotometer (HP 8453). 

### 4.3. Mass Spectrometry Analysis of Recombinant Hemoglobins

Mass spectrometry (MS) experiments were performed with 60 μM (heme) native HbA, HbS (βE6V), Hb Providence (K82D). We previously reported similar MS analysis on HbS Providence (βE6V/K82D) and cross-linked Providence (rHb0.1/βK82D) (see Table 1) [15,16]. To study the impact of the βK82D mutation on H_2_O_2_-mediated βCys93 amino acid oxidation, the above proteins were incubated in PBS buffer that was treated with increasing (2.5×, 5× and 10×) molar excess of H_2_O_2_ per heme. All oxidation reactions were carried out in phosphate buffer saline (PBS), pH 7.4, at an ambient temperature for 30 min. A total of 1 µL of 1 unit/µL catalase was added to remove excess H_2_O_2_ to quench oxidation in the Hb samples.

### 4.4. LC/MS/MS Analysis 

All Hb samples were tryptically digested, desalted, and analyzed by mass spectrometry, as previously described [47]. Briefly, tryptic peptides were analyzed by reverse phase liquid chromatography mass spectrometry (RP LC/MS/MS) using an Easy nLC II Proxeon nanoflow HPLC system coupled online to a Q-Exactive Orbitrap mass spectrometer (Thermo Fisher Scientific, Waltham, MA, USA). Data were acquired using a top 10 method (for 60 min), dynamically choosing the most abundant precursors (scanned at 400–2000 *m*/*z*), from the survey scans for HCD fragmentation. 

All samples were searched against the Swiss-Prot Human database (release 2014_03, containing 542,782 sequence entries) supplemented with the porcine trypsin sequence, using the Mascot (version 2.4) search engine (Matrix Sciences, London, UK), as described previously [47]. Variable modifications including cysteine trioxidation (+48 Da), methionine oxidation (+16) were included for identifying the oxidative modifications. Carbamidomethylation of cysteine (+57) was also included as a static modification for all unoxidized cysteines. Mascot output files were analyzed using the software Scaffold 4.2.0 (Proteome Software Inc., Portland, OR, USA). In addition, αHb peptide identifications were accepted if they could be established at greater than 99.9% probability and contained at least 2 identified peptides. Probabilities were assigned by the Protein Prophet algorithm [48]. Each peptide was further validated by retention time reproducibility.

### 4.5. Quantitative MS Analysis 

All quantitative experiments were performed in triplicates and the standard deviations were obtained by averaging relative abundance data from three different experiments. Extracted ion chromatograms (XICs) were generated from the most abundant monoisotopic peak of isotopic profiles representing the charged states of each peptide (oxidized and unoxidized). To construct XICs, the Xcalibur (version 2.4) software (Thermo Fisher Scientific, Waltham, MA, USA) was used with a designated mass tolerance of 0.01 Da, and mass precision was set to three decimals. For relative quantification, the ratio of each isoform was calculated based on the sum of the XIC peak area from all forms, which was normalized to 100% and included all charge states and versions that result from different cleavage sites (more details can be found in [17]).

### 4.6. Endothelial Cell Culture

Cryopreserved human pulmonary artery endothelial cells (HPAEC) (Thermo Fisher Scientific, Waltham, MA, USA) were cultured in a specially formulated media (Medium 200) containing 2% fetal bovine serum (FBS), supplemented with Low Serum Growth Supplement (LSGS) (Thermo Fisher Scientific, Waltham, MA, USA). For all experiments, HPAEC were used between passages 5–10. 

### 4.7. Treatment of HPAECs with Hemoglobin Variants

HPAECs were grown to 80–90% confluency in complete media, before any treatments. Cells were then serum starved for 12 h in an FBS-free medium composed of Medium 200, supplemented with all other components of the LSGS Kit except FBS (Thermo Fisher Scientific, Waltham, MA, USA). The cells were exposed to various Hb variants at equimolar concentrations (100 µM) in their ferrous form (HbFe^2+^) for 24 h in an FBS-free growth media. After incubation, the cells were washed with pre-warmed phosphate buffered saline (PBS) for three times, to remove any residual Hb proteins in the media. Cells were then lysed with RIPA lysis and extraction buffer (Thermo Fisher Scientific, Waltham, MA, USA) containing the protease inhibitor for further studies. 

### 4.8. Gel Electrophoresis and Immunoblotting and Fluorescence Microscopy

HPAEC lysate proteins were resolved by SDS–PAGE using precast 4–20% NuPAGE bis-tris gels (Thermo Fisher Scientific, Waltham, MA, USA) and then transferred to nitrocellulose membranes (BioRad, Hercules, CA, USA), by the standard immunoblotting technique. Nitrocellulose membranes were processed with different specific primary antibodies and appropriate secondary antibodies. Mouse monoclonal antibodies against HO-1 (ab13248), beta actin for loading control (ab8226), rabbit polyclonal antibodies against light chain of ferritin light chain (ab69090), TLR4 (ab13556), VCAM1 (ab134047) were purchased from Abcam (Cambridge, MA, USA). Appropriate HRP-conjugated goat anti-mouse IgG (ab97040) and anti-rabbit IgG (ab205718) secondary antibodies were also obtained from Abcam (Cambridge, MA, USA). 

We also employed immunohistochemistry to detect and visualize intra-cellular HO-1 expression in Hb-treated endothelial cells. HPAECs were grown up to 50% confluency on collagen I coated coverslips placed on 12-well plates and then incubated with various Hb mutants for 12 h. Following incubation, the cells were thoroughly washed in PBS to remove traces of Hb proteins. Immunohistochemistry was done as described earlier, following fixation of cells in 4% paraformaldehyde solution and permeabilization with PBS containing 1% BSA and 0.1% Triton X-100 [18]. HO-1 expression in the cells were detected with mouse anti-HO-1 (ab13248) antibody and visualized under a fluorescence microscope (EVOS, Thermo Fisher Scientific, Waltham, MA, USA), using a secondary goat anti-mouse AlexaFluor595 conjugated antibody and mounting with Prolong-Gold antifade mountant containing DAPI (Thermo Fisher Scientific, Waltham, MA, USA).

### 4.9. Endothelial Bioenergetic and Glycolytic Flux Measurements

Cellular oxygen consumption and glycolytic rate were assessed in real-time, using an Agilent-Seahorse XF24 Extracellular Flux analyzer (Agilent, Santa Clara, CA, USA), as described before [21]. In brief, 80,000 cells/well were cultured in collagen I-coated 24-well XF-V7 cell culture plate (Agilent, Santa Clara, CA, USA) for 24 h. Cells were then serum-starved for 12 h, prior to Hb treatment for 24 h. Following the incubation, the Hb containing media was gently washed once with PBS and replaced with 500 µL of the XF-assay media (Agilent, Santa Clara, CA, USA), supplemented with 10 mM glucose, 5 mM pyruvate, and 2 mM glutamate. Mitochondrial oxygen consumption rate (OCR) was assessed under different bioenergetic states, e.g., coupled, uncoupled, and inhibited states, created by automated sequential injections of oligomycin (1 µM), carbonyl cyanide-p-trifluoro-methoxyphenylhydrazone (FCCP, 1 µM), and a combination of mitochondrial inhibitors (rotenone, 1 µM and antimycin A, 1 µM), respectively [21]. Similarly, endothelial glycolytic capacity was assessed by measuring the extracellular acidification rate (ECAR). For the ECAR experiments, glucose-free XF-assay media were used. Real-time glycolytic profile was obtained by sequential addition of glucose (10 mM), oligomycin (1 µM), and glycolytic inhibitor 2-deoxyglucose (2-DG, 100 mM) to the wells.

The OCR and ECAR values were plotted using the XF24 software, version 1.8. To eliminate any background OCR or ECAR, few blank wells with Hb variants were also run. Various bioenergetic and glycolytic parameters were calculated following the manufacturer’s protocol and as described earlier [49]. In brief, basal respiration was considered as the difference between maximum OCR obtained before oligomycin addition and non-mitochondrial OCR obtained after rotenone/antimycin A, whereas the maximal respiration was the difference between maximum OCR induced by FCCP and non-mitochondrial OCR. Similarly, glycolysis was considered as the maximum ECAR obtained after addition of glucose and the glycolytic capacity was the maximum ECAR achieved by oligomycin addition that shuts down ATP generation oxidative phosphorylation. 

### 4.10. Statistical Analysis

Plotting of data and statistical calculations were done with GraphPad Prism 7 software. All values are expressed as mean ± SEM. A *p*-value of <0.05 was considered to be statistically significant. The difference between two means were compared using unpaired Student’s *t*-test.

## Figures and Tables

**Figure 1 ijms-21-09453-f001:**
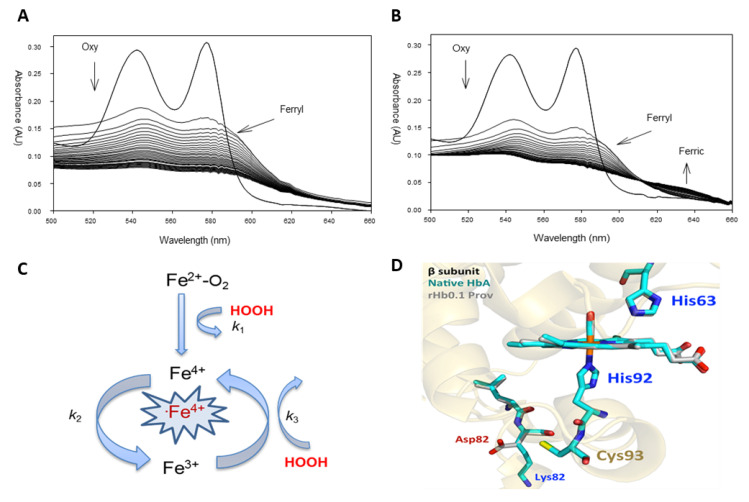
Pseudoperoxidase activity of hemoglobin Providence. (**A**) Spectral changes in wild-type hemoglobin A (β82K) (arrows pointing downward indicate the direction of the reaction and the conversion of the oxy/ferrous (two main peaks; 541 and 577 nm) to ferryl heme (two main peaks; 545 and 580 nm)). (**B**) Hemoglobin Providence (βK82D) during oxidation by hydrogen peroxide. (**C**) A model represents the pseudoperoxidase activity of hemoglobin. (**D**) Comparison of the active sites, EF corner, F-helix, and C-terminus of β subunits in HbA (cyan, 2DN3) and rHb0.1/βK82D (rHb0.1 Prov) (gray, 5SW7). The drawing was made by MOL Molecular Graphics System, version 2.0 (Schrodinger, LLC) (New York, NY, USA). Panels A, B, and C were reprinted with permission from [14,17].

**Figure 2 ijms-21-09453-f002:**
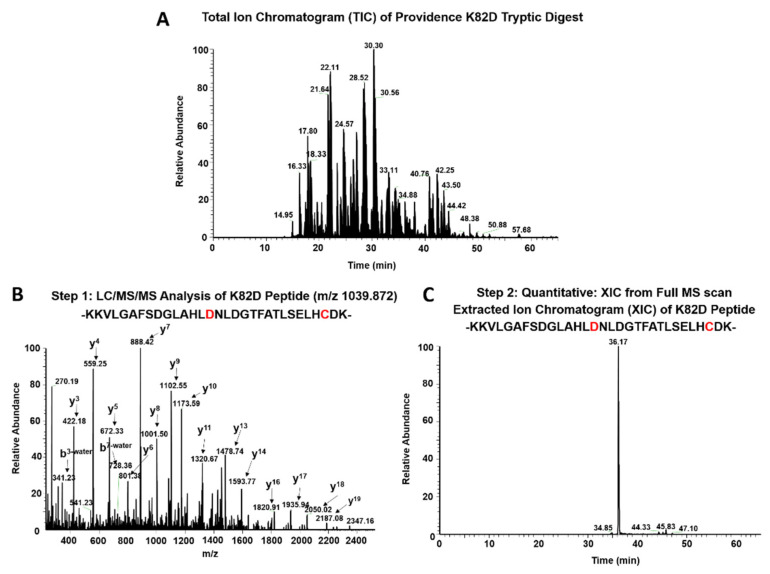
Quantitative LC/MS/MS analysis of hemoglobin Providence. (**A**) Total Ion Chromatogram (TIC) collected during a 60-min mobile phase gradient separation of tryptically digested Hb Providence peptides (after treatment with 10× H_2_O_2_), using C18 reverse phase chromatography. (**B**) LC/MS/MS fragmentation spectra of the βK82D containing +3 peptide KKVLGAFSDGLAHLDNLDGTFATLSELHCDK (1039.872 *m*/*z*), which eluted at 36.11 min. Step 1: Database searches from LC/MS/MS analysis were utilized to identify all oxidized and unoxidized C93 containing βK82D peptides. The y an b ion series from this spectrum provided (representing the βK82D peptide with unoxidized C93) sequence information that was used to confirm the peptide identify prior to quantification. (**C**) Extracted ion Chromatogram (XIC) of the same peptide identified from MS/MS spectra represented in panel 2. Step 2: After database searches were used to identify all βK82D peptides containing oxidized and unoxidized C93, extracted ion chromatograms (XICs) from full MS scans were used to quantify the relative abundance of oxidized βK82D.

**Figure 3 ijms-21-09453-f003:**
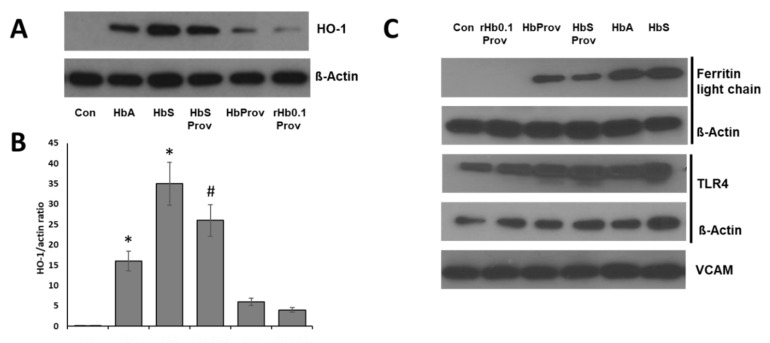
Effect of hemoglobins carrying Providence mutation on human pulmonary arterial endothelial cells. HPAECs were exposed to ferrous forms of either HbA, Providence (βK82D), HbS, HbS Providence (βE6V/βK82D), or crosslinked Providence (rHb0.1/βK82D), at equimolar concentration (100 µM) for 24 h. (**A**,**C**) Cell lysates were immunoblotted with primary antibodies against HO-1, ferritin light chain (L-ferritin), toll-like receptor 4 (TLR4), and vascular cell adhesion molecule 1 (VCAM). Equal loading was confirmed by re-probing the blots against β-actin. All immunoblot panels shown are representatives of three separate independent experiments. (**B**) Densitometric analysis was done for the HO-1 expression vs. corresponding β-actin levels and the values represent the average ratio of band intensities (HO-1:β-actin), *n* = 3. * *p* < 0.05 vs. untreated control; ^#^
*p* < 0.05 vs. corresponding HbS.

**Figure 4 ijms-21-09453-f004:**
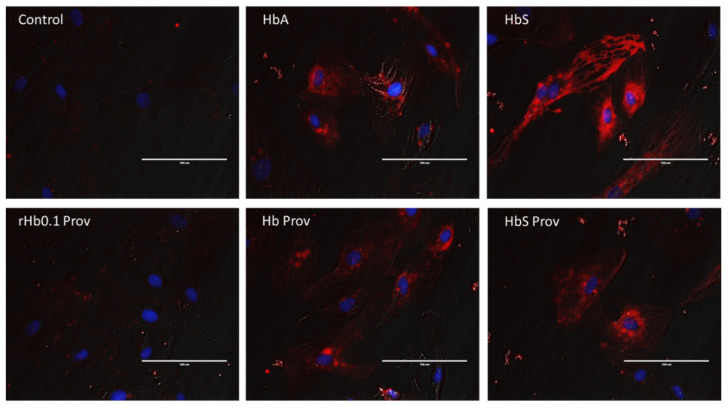
Immunohistochemistry of HO-1 expression in HPAECs. HPAECs were grown on coverslips and incubated with or without HbA, Providence (βK82D), HbS, HbS Providence (βE6V/βK82D), or crosslinked Providence (rHb0.1/βK82D), at equimolar concentration (100 µM), for 12 h. Immunohistochemistry was done in paraformaldehyde fixed HPAECs, using antibody against HO-1, as described in the Methods section. Representative fluorescence microscopic images of HPAEC shows HO-1 expression (red fluorescence, AlexaFluor 595). Cell nuclei appear as blue (4′,6-diamidino-2-phenylindole, DAPI). Fluorescence images were merged onto the corresponding phase contrast views to indicate cellular structures. All images are representative of several similar fields obtained from two separate experiments. White line indicates 100 µm.

**Figure 5 ijms-21-09453-f005:**
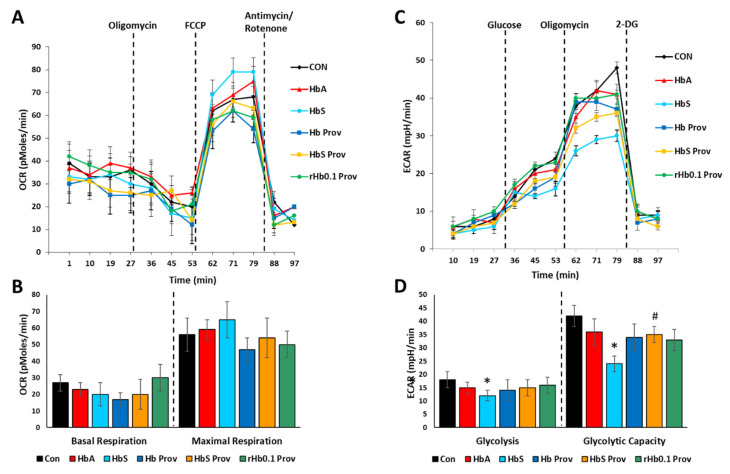
Effect of mutant hemoglobin proteins on pulmonary endothelial bioenergetics. HPAECs were exposed to ferrous forms of HbA, Providence (βK82D), HbS, HbS Providence (βE6V/βK82D), or crosslinked Providence (rHb0.1/βK82D) at an equimolar concentration (100 µM) for 24 h. Mitochondrial oxygen consumption rates (OCR) and extracellular acidification rates (ECAR) were measured by Agilent Seahorse (XF24) extracellular flux analyzer in real time. (**A**) Bioenergetic profile indicating average OCR values from four similar wells treated with Hb mutants. (**B**) Bar diagrams showing basal and maximal respiration (OCR) calculated from the OCR plots (**A**), following exposure to Hb (N = 4). (**C**) Glycolytic lactate production was measured as ECAR and plotted as the average of four similar wells treated with Hb mutants. (**D**) Basal glycolysis rate and glycolytic capacity were calculated from the ECAR plots of HPAEC, following exposure to Hb (N = 4). Representative OCR and ECAR plots were obtained from an individual set of experiment repeated three times. * *p* < 0.05 vs. the untreated control; ^#^
*p* < 0.05 vs. corresponding HbS.

**Table 1 ijms-21-09453-t001:** Oxidation parameters of the hemoglobin Providence constructs.

Treatment Conditions HPAECs	Cys93 Oxidation	Cys93 Oxidation	Cys93 Oxidation
	2.5:1	5:1	10:1
	(H_2_O_2_:heme)	(H_2_O_2_:heme)	(H_2_O_2_:heme)
HbA [18]	21 ± 4.2%	31 ± 2.8%	58 ± 3.8%
HbS (βE6V) [18]	32.9 ± 2.7%	64 ± 5.0%	67% ± 6.8
Hb Prov (βK82D)	6.8 ± 3.1%	11.6 ± 0.8%	9.5 ± 4.3%
HbS Prov (βE6V/βK82D) [16]	11.9 ± 2.0%	N/A	22 ± 1.3%
Crosslinked Providence (rHb0.1 Prov) [15]	Below detection	Below detection	2.7 ± 0.2%

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
