# Peer review of "The Providence Mutation (βK82D) in Human Hemoglobin Substantially Reduces βCysteine 93 Oxidation and Oxidative Stress in Endothelial Cells"

_ijms, 2020, doi:10.3390/ijms21249453_

Round 1

Reviewer 1 Report

See attached file

Author Response

The work by Alayash and coworkers indeed is interesting, even though it is only an extension of previous studies, which demonstrated convincingly that the substitution of Lys β82 by Asp significantly decreased the oxidation of Cys β93 to cysteic acid; this protection then results in a reduced heme release and oxidative damage. As usual for this group, experiments are very well planned and performed, so that I have no criticism about them. However, I have some critical considerations about the conclusions. A first point is that actually, as it is evident from Figs. 1A and 1B, the oxidation rate is not very different between HbA0 and the Providence mutant; what is different (and it is very important!) is that in the case of the mutant is the formation of the ferryl species to be impaired! Therefore, it is the formation of the ferryl radical to oxidize Cys β93 to cysteic acid and the mutant effect is to avoid this harmful intermediate.

RESPONSE: We agree with this reviewer that the rates of the “initial” oxidation (ferrous to ferryl) in HbA0 and the Providence mutations are indeed very close. What is different however, is that the ferryl species once formed remains in HbA0 solutions whereas in the case of the Prov mutations the ferryl is rapidly autoreduced back to the ferric species completing an efficient pseudoperoxidative cycle and therefore less damage to βCys93.  Figs.1A and 1B therefore emphasis these differences (qualitatively) between the proteins as have been shown by our group previously (Strader et al., Biochem J. 2017; Meng et al., JBC, 2019).

In this respect, this paper does not add any information about the structural basis of this different behaviour and what is the mechanistic role of the Asp 82 mutation in this effect, which should be instead the crucial information to avoid this harmful cycle through the ferryl radical. As a matter of fact, this paper is only showing that endothelial cells are affected by the release of heme and oxidative by-products due to the xoidation of Cys β93 to cysteic acid, which is a very limited novel information. Although I realize that it would change too much the scope of the paper, I’d suggest authors to more deeply investigate the mechanism through which the Asp substitution at position 82 (i.e., in the central dyad axis of β chains) avoids the formation of the ferryl species.

RESPONSE:  Mechanistic link between βCys93 oxidation in each protein with HO-1 and other oxidative stress markers in endothelial cells is the first step towards establishing these important pathways in animal models/human studies (Alayash, AI, J Lab Investig, 2020). We have indeed shown recently that posttranslational modifications including βCys93 oxidation do occur in hemolysates from SCD animals and humans (Strader et al., J Clin Invest, 2018; Jana et al., Sci Rep, 2020). Establishing the underlying biochemical mechanism that fully explain the contribution of Asp82 to the overall oxidative stability of the mutant Hbs is beyond the scope of this paper. We did however, previously offered a potential mechanism based on crystal structure of the HbProv (Figure 1) (Strader et al., Biochem J, 2017) briefly summarized in the discussion section. However, the contribution of Asp at position 82 to the autoreduction of ferryl Hb (pseudoperoxidative activity) as the reviewer suggests may indeed play additional role in this extraordinary stability of these mutant Hbs. Enzymatic kinetic as well as structural studies may thus provide more additional mechanistic insights as to why Prov mutations are evolutionary stable proteins and are underway. We have now added this statement as per this reviewer’s comment in the discussion section.

Hydrogen peroxide dependent oxidation reactions of the oxy form of the βK82D mutant may also reveal additional mechanistic insights into differences in the kinetics of ferryl heme formation and reduction and the potential role of Asp82 in the protein redox transitions.   

A second weak point of the paper is that the suggestion of targeting gene-editing with this substitution to ameliorate the oxidative damages in sickle cell disease appears very unrealistic also in view of the lack of information on the effect of the mutation on the oxygen delivery to tissues in animal models. Therefore, they should underline this weakness in the conclusions in a more clearcut way.

RESPONSE: Current genetic engineering technologies are aimed at expressing non-sickle cell Hbs within the SCD patient’s RBCs (~%30-40). Most commonly overexpressed proteins are HbF or naturally occurring mutant Hbs (e.g. βT87Q) (Negre O et al., Hum Gene Ther 27:148-165, 2016). The advantage of diluting out HbS within RBCs is to reduce the degree of polymerization/sickling due to the fact that these mutants generally don’t co-polymerize with HbS in addition they provide a modest left shift in overall oxygen equilibrium curve of RBCs.  Hb Prov as we have shown enhances not only oxidative stability, but it increases delay time considerably thus it may provide a superior benefit over other non-S Hbs and (if expressed in SS RBCs) and it may aid in reducing vaso-occlusion in the microcirculation. We are currently working with other colleagues from other institutions to design relevant animal studies that enable us to test these unique properties of Hb Prov mutants in sickle cell mouse model.

Reviewer 2 Report

The present manuscript describes very potentially impactful work demonstrating effects of an amino acid mutation on the stability of hemoglobin (Hb) toward H2O2-mediated oxidation, and the effects of the various H2O2-treated proteins, on endothelial cells that then mount a response, recorded as HO-1 or ferritin expression.  The main finding is very important, that this naturally-occurring mutation, K82D, confers stability toward oxidation of the (beta)Cys93 thiol group in the context of the sickle cell mutation (E6V) or a cross-linked form of Hb.  The work is, on the whole, very well conducted, explained and interpreted.  I do find, however, that there are a few issues that need to be resolved for the manuscript to be acceptable for publication.

Major

The authors indicate that there are two versions of the Cys93 peptide, oxidized or unoxidized.  Oxidized refers to the sulfonic acid form (aka cysteic acid).  Since there are also intermediate states that are undoubtedly formed (sulfenic and sulfinic acid states), it is important to mention whether or not these are ever observed or considered.  This may be addressed in other papers that are referenced, but the reader should not have to go look those up for this information, and the interpretation of considering just these two states should be justified.  Perhaps once sulfenic or sulfinic acid species are formed in the protein, they are highly susceptible to further air-mediated oxidation during sample workup including tryptic digestion (although the intact proteins used to treat the HPAECs may still have the intermediate oxidized species, which can be stable or meta-stable).  But even the abstract refers to irreversible oxidation to cysteic acid as if no intermediates are formed in the intact protein.  This needs to be addressed by the authors.  Use of a trapping agent like dimedone to seek evidence of a sulfenic acid intermediate could help with this if not already tried.

Minor

Trying to follow the spectral changes shown in Fig. 1A and B and described in the text, I find the labeling of “Oxy” and its arrow a bit confusing.  Could the arrow simply be turned to point to the nearby most pronounced peaks?  In the text and legend, there is no mention of “Oxy”, but the text mentions “ferrous HbAo”.  This could be made much clearer for the reader.

In Fig. 1C, a hypothetical model for the cycling (pseudoperoxidase) mechanism is included, along with rate constants k(2) and k(3).  I find that the text of the paper does not follow this to a conclusion, however, other than rather general statements.  Can’t k(2) and k(3) be determined (or at least estimated) on the basis of the spectral data (perhaps k(1) as well)?  It would be helpful for the authors to then state their conclusions in terms of these kinetic rate constants.

Figure 4C on line 228 should be Figure 5C.

Differences in maximal respiration (Fig. 5A and B, lines 223-226) and basal glycolysis (Fig. 5D, lines 229-230) are described in the text, but refer to outputs that are not marked in the plots as statistically significantly different from the control.  It is not appropriate to describe these “differences” unless minimally the statement is included that these did not reach statistical significance.  The next to last sentence of the discussion (lines 334-336) also seems to refer to some changes that were not actually observed or significant.

The term “reversible” on line 260 is probably meant to be “irreversible” since it refers to cysteic acid.

Line 274, “resistant” should be “resistance”. There are some other minor grammatical and typographical mistakes, as well.

Treatment of cells with iodoacetamide, presumably during lysis, can be inferred from the detection of “non-oxidized” hemoglobin peptides as carboxamidomethylated cysteine containing peptides, but I do not see in the methods where this is mentioned.

Author Response

The authors indicate that there are two versions of the Cys93 peptide, oxidized or unoxidized.  Oxidized refers to the sulfonic acid form (aka cysteic acid).  Since there are also intermediate states that are undoubtedly formed (sulfenic and sulfinic acid states), it is important to mention whether or not these are ever observed or considered.  This may be addressed in other papers that are referenced, but the reader should not have to go look those up for this information, and the interpretation of considering just these two states should be justified.  Perhaps once sulfenic or sulfinic acid species are formed in the protein, they are highly susceptible to further air-mediated oxidation during sample workup including tryptic digestion (although the intact proteins used to treat the HPAECs may still have the intermediate oxidized species, which can be stable or meta-stable).  But even the abstract refers to irreversible oxidation to cysteic acid as if no intermediates are formed in the intact protein.  This needs to be addressed by the authors.  Use of a trapping agent like dimedone to seek evidence of a sulfenic acid intermediate could help with this if not already tried.

RESPONSE:  We thank the reviewer for making this point.  Cys93 has been shown to be an important endpoint for free radical-induced protein oxidation within the β Hb subunit. Thus, the formation of this oxidation moiety to cysteic acid depends on the presence of globin centered radicals and ferryl ions.  In a previous paper (Kassa et al 2015 JBC., vol. 290 27939-27958) where we studied HbS oxidative stability directly, we used the trapping agent DMPO to show that globin radicals and ferryl ions were more prevalent in HbS than HbA.  The exposure of Hb to H2O2 has previously been shown to initially produce a porphyrin cation radical (and ferryl ion) that oxidizes cysteinyl amino acids.  DMPO labeling allowed us to show that HbS was oxidatively more reactive than HbA by virtue of having a larger abundance of the oxidized Fe+4 species. While we agree with the reviewer that the other intermediates occur, our previous characterization of Hb oxidation has shown that trioxidation to cysteic 93 is a prevalent reaction that occurs as a response to H2O2 and is therefore a good reporter of ascertaining the oxidative stability of a Hb variant relative HbA.

To address the authors comments we have changed the manuscript text to clarify this point.

“We have previously used LC/MS/MS analysis to determine that Hb toxicity is linked to irreversible βC93 oxidation to cysteic acid at the amino acid side chain. Comparing βC93 oxidation of Hb variants with HbA has served as an effective method for evaluating Hb oxidative stability [17]”.

Changed to:

We have previously used DMPO labeling studies [18] and LC/MS/MS analysis to show that Hb toxicity is linked to irreversible βC93 oxidation to cysteic acid by virtue of ferryl ions and globin radicals.  While other intermediates may occur, comparing βC93 oxidation to cysteic acid has been an effective method for evaluating Hb oxidative stability relative to HbA.

Minor

Trying to follow the spectral changes shown in Fig. 1A and B and described in the text, I find the labeling of “Oxy” and its arrow a bit confusing.  Could the arrow simply be turned to point to the nearby most pronounced peaks?  In the text and legend, there is no mention of “Oxy”, but the text mentions “ferrous HbAo”.  This could be made much clearer for the reader.

RESPONSE: The arrow pointing downward indicates the direction of the reaction with time as oxy/ferrous is transformed to ferryl. We have now added in the legend the characteristic wavelengths for the ferrous/oxy (that we used interchangeably) and that of the ferryl. We thank the reviewer for pointing out this confusion. We have now added the following clarifying statement in the legend:

  1. Spectral changes in wild-type hemoglobin A (β82K) (Arrows pointing downward indicate the direction of the reaction and the conversion of the oxy/ferrous (two main peaks; 541 and 577nm) to ferryl heme (two main peaks; 545 and 580 nm).

We have also indicated in parenthesis that we use ferrous Hb and ferric Hb as oxyHb and metHb respectively throughout the text interchangeably.

In Fig. 1C, a hypothetical model for the cycling (pseudoperoxidase) mechanism is included, along with rate constants k(2) and k(3).  I find that the text of the paper does not follow this to a conclusion, however, other than rather general statements.  Can’t k(2) and k(3) be determined (or at least estimated) on the basis of the spectral data (perhaps k(1) as well)?  It would be helpful for the authors to then state their conclusions in terms of these kinetic rate constants.

RESPONSE: the original model we have described earlier was based on analyzing very complex spectral data generated from the reaction of increasing concentrations of peroxide with recombinant myoglobins and several alpha or beta crosslinked Hbs. A mechanism, based on singular value decomposition (SVD) and nonlinear least squares kinetic analysis of time-base spectral data was subsequently described (Alayash J Biol Chem. 1999 Jan 22;274(4):2029-37; Alayash and Cashon, Arch Biochem Biophys. 1995 Jan 10;316(1):461-9). This working model in its simplest form is presented in Figure 1C. This will enable the reader to appreciate the complex transition among Hb redox forms. Similar complex kinetic studies on the HbProv mutants we believe is beyond the scope of this paper. We have however, previously reported the rate of ferryl autoreduction (k2) and ascorbate mediated reduction for rHbSβK82D (Meng et al., JBC, 2019) and in separate study we provided kinetic data on oxidation reactions and heme loss kinetics of rHb0.1βK82D (Strader et al., Biochem J, 2017).

Figure 4C on line 228 should be Figure 5C.

RESPONSE: We have corrected the figure in the text. We thank the reviewer for spotting this error

Differences in maximal respiration (Fig. 5A and B, lines 223-226) and basal glycolysis (Fig. 5D, lines 229-230) are described in the text, but refer to outputs that are not marked in the plots as statistically significantly different from the control.  It is not appropriate to describe these “differences” unless minimally the statement is included that these did not reach statistical significance.  The next to last sentence of the discussion (lines 334-336) also seems to refer to some changes that were not actually observed or significant.

RESPONSE: We thank the reviewer for pointing these errors. We have corrected the sentence as follows:

However, βE6V treated cells showed mild but non-significant uncoupling over the untreated control cells, perhaps mediated by TLR4 activation and induction of HO-1 expression [21].

For basal glycolysis we have seen ~30% loss by HbS as mentioned in the text. We have now indicated this statistically significant change in the figure 5D by placing an appropriate symbol (*).

We have also corrected the last paragraph of the discussion by emphasizing the recovery of endothelial glycolytic parameters seen in our study replacing ‘mitochondrial respiration’. Corrected sentences are as follows:

In summary, we have set out to examine the mechanistic link between irreversible oxidation of a key amino acid in the β subunit of Hb, Cys93, oxidative toxicity, cellular redox homeostasis and energy metabolism in cultured endothelial cells.

Oxidative modifications triggered by βCys93 oxidation to cysteic acid and subsequent heme loss from normal and sickle cell Hbs caused impairment of pulmonary endothelial glycolysis that were effectively blunted by βK82D mutation into HbS and crosslinked human Hb.

The term “reversible” on line 260 is probably meant to be “irreversible” since it refers to cysteic acid.

Line 274, “resistant” should be “resistance”. There are some other minor grammatical and typographical mistakes, as well.

RESPONSE: We have corrected the errors. We thank the reviewer for spotting this.

Treatment of cells with iodoacetamide, presumably during lysis, can be inferred from the detection of “non-oxidized” hemoglobin peptides as carboxamidomethylated cysteine containing peptides, but I do not see in the methods where this is mentioned.

RESPONSE: We thank the reviewer for bringing up this point.  We routinely use reduction and carbamidomethylation with iodoacetamide (IAM) in prepping samples for MS analysis.  In this manuscript we referenced a previous paper with regard to preparing the samples for data analysis.  We did however, include carbamidomethylation of cysteine (+57) as a static modification for all cysteine containing peptides.

Round 2

Reviewer 1 Report

Although the authors did not answer completely to my main point (a structural explanation of why the mutation substantially eliminates the ferryl formation), they answered satisfactorily to all my other concerns. I also understand that a substantiated answer to my main point is not simple and it is perhaps outside the scope of this paper.

Therefore, I consider this revised version satisfactory, deserving the publication.